# Synthesis of Geared Planar Linkage Mechanisms through the Segmentation of Multiloop Mechanisms into Discrete Chains

**Sean Mather \* and Arthur Erdman**

Mechanical Engineering, University of Minnesota, 111 Church Street SE, Minneapolis, MN 55455, USA;
agerdman@umn.edu
\* Correspondence: mathe587@umn.edu

**Abstract:** Gears are foundational tools used to transmit or modify mechanical energy or motion. Implementing gears into planar linkage mechanisms is less common but can be a similarly valuable technique that takes advantage of the high efficiency of gears while producing complex and precise motions. While recent numerical methods for designing these geared planar linkage mechanisms (GPLMs) have proliferated in the literature, analytical approaches have their merits and have received less attention. Here, an analytical alternative is presented as a modification of the complex-number loop-based synthesis method for designing multiloop mechanisms. Some of the base topologies for geared dyad, triad, and quadriad chains are presented, along with a numerical example demonstrating the solution procedure's effectiveness.

**Keywords:** geared planar linkage mechanism; kinematic synthesis

## 1. Introduction

The earliest known example of a geared device created is believed to date back as far as the year 205 B.C. Called the Antikythera mechanism, this device was used to predict the behaviors of celestial bodies [1,2]. Later, more examples of geared mechanisms were identified dated to the fourth century [3] (p. 54). Despite their long history, gears remain prominent in mechanical systems today as a central method of power transmission due largely to their reliability and efficiency. The earliest known examples of the incorporation of gears in mechanical linkages date back to the 1700s, per Shirkhadoie [4]. In more recent times, research in this area has approached the problem of geared linkage synthesis through two main avenues. More than 20 years ago, research in geared linkage mechanisms primarily focused on finding analytical solutions, while more recently, work in this field emphasized using numerical methods to derive novel mechanisms, especially complex path-generating mechanisms.

An example application of a geared planar linkage mechanism (GPLM) is shown in Figure 1. This mechanism harnesses wave energy to generate electricity by driving a piezoelectric film [5]. An additional example is provided in Figure 2, depicting a geared five-bar 'pick and place' mechanism in two positions, with an added driving dyad to restrict the motion range. The calculations used to synthesize the mechanism in Figure 2 are shown at the end of the manuscript. This paper is focused on mechanisms with combinations of gear connections and linkage members. The epicyclic gear train in Figure 3 is a representative example of a gear train style that is not the focus of this paper. Note that, like many compound gear-trains, this may be deployed in either a single or two-degree-of-freedom application, depending on whether the ring gear is fixed to the ground.

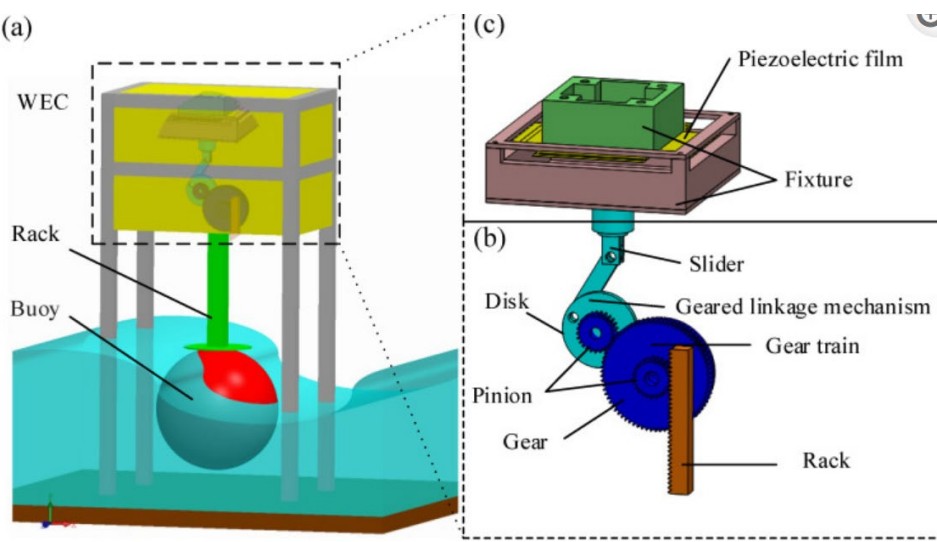

**Figure 1.** A mechanism designed to harness the energy of waves as electrical power [5].

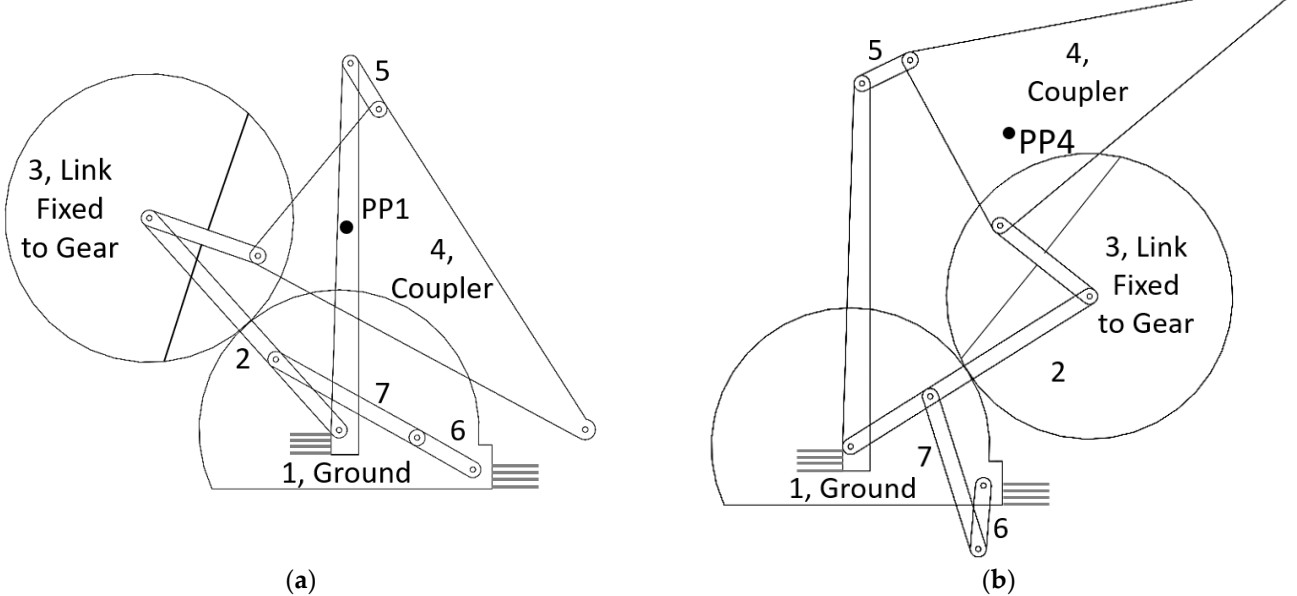

(**a**)                                              (**b**)

**Figure 2.** An example of a 7-bar-geared linkage mechanism depicted in two positions, synthesized for motion generation with prescribed timing. The gears are affixed to link one (ground) and link three. The mechanism was designed as a base five-bar-geared mechanism, and a driving dyad was added to constrain the motion of the rocker input. By driving link 6, this dyad protects the mechanism against possible branch defects at the fringes of the rocker's range of motion. (**a**) a depiction of the mechanism in its first position. (**b**) a depiction of the mechanism in its fourth position.

This manuscript is divided into four main sections. Here, in Section 1, this manuscript and its goals have been briefly described. In Section 2, the existing synthesis in the literature for both analytical and numerical methods is summarized. Additionally, a handful of existing geared linkage mechanism examples are shown to illustrate some of the advantages of gears and how the inclusion of gears in linkage mechanisms can produce complex motions. In Section 3, a uniform analytical synthesis method, amenable to numerical variations, is presented, along with several examples of topological chains that pair well with the present method. Finally, in Section 4, a numerical example is provided to demonstrate the synthesis procedure, and the results are compared against a pair of four-bar linkages to give evidence of the efficacy of GPLMs.

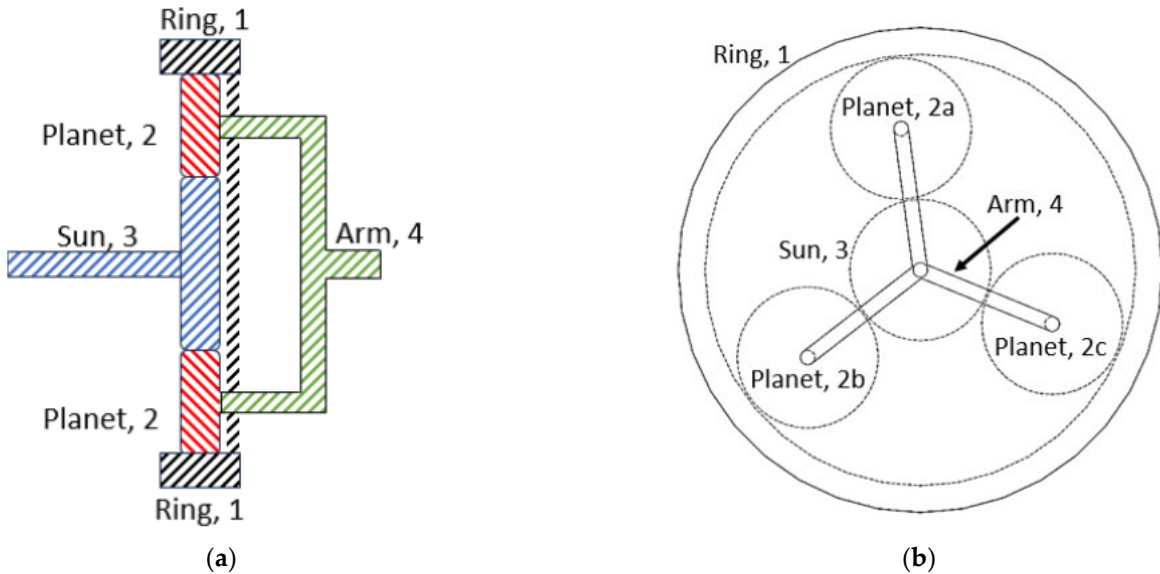

**Figure 3.** Traditional gear trains also abound in the literature. A typical one is shown here. Though gear trains are not the emphasis of this paper, a planetary gear train like this one may be a useful addition to modulate the torque of a geared planar linkage mechanism. (**a**) A side view of an epicyclic gear train. Note that while both the planets are labeled link 2, they are separate elements, as seen in Figure 3b. (**b**) A front view of an epicyclic gear train.

## 2. Literature Review

Some of the earliest analytical analyses of geared linkages in the modern era are attributed to Dr. Ferdinand Freudenstein, who identified the coupler curve equation for an arbitrary five-bar geared linkage mechanism [6,7]. Rao and Sandor expanded Freudenstein's equations for synthesizing a four-bar linkage to accommodate the synthesis of GPLMs [8]. Sandor and Erdman demonstrated analytical geared linkage synthesis for a function generation task [9]. Flugrad and Starns used continuation methods to synthesize seven prescribed-position-geared path generators [10]. Here, prescribed positions refer to the designer (or problem)-defined positions in the plane that the path tracer point, a point on the mechanism, must pass through. In a 'motion generation' problem, the prescribed positions are not just points in the plane but also include the angle of the moving plane at each position. Sandhya et al. synthesized a geared five-bar mechanism that emulated the path of a hummingbird's wings based on parametric curve approximation. Parlaktaş demonstrated the synthesis of geared adjustable stroke mechanisms through a few different approaches, including inflection circle analysis [11]. Tso and Liang demonstrated analytical synthesis techniques as they divided a geared nine-bar mechanical press mechanism into two distinct mechanisms connected via a floating link [12]. Modler et al. did some type of synthesis work based on complex numbers for geared linkages with linear actuators designed for the synthesis of function generator mechanisms [13].

In addition to synthesis, methods for analyzing or improving geared mechanisms have also been well investigated. Synthesizing a geared mechanism that follows a particular path was also demonstrated by Zhang, who showed how the properties of geared five-bar mechanisms could be improved using an atlas, or compilation, of known coupler curves [14]. Soong discussed methods of analyzing geared linkage mechanisms [15]. Similarly, Ting provides a methodology similar to the Grashof criteria for four-bar linkages, which allows a designer to anticipate the 'rotability' of a geared five-bar mechanism given the dimensions of the mechanism and some phase angle conditions [16]. Pennock and Sankaranarayanan expand on Pennock's earlier work [17] to demonstrate a graphical technique for finding the instant centers of geared mechanisms [18].

In the last twenty years or so, numerical approaches have been favored by researchers. Jacek Buśkiewicz used evolutionary algorithms to derive geared five-bar path-generating linkages with gears [19]. Sang Min Han, Suh In Kim & Yoon Young Kim discussed a numerical method known as spring-block modeling for synthesizing path-generating mechanisms without prescribed timing [20]. Yoon Young Kim, Seok Won Kang, and Neung Hwan Yim expanded this work on spring-block modeling to accommodate GPLMs, again applying a topology-optimizing path generation algorithm [21]. Others mix methods, using analytical approaches to synthesize a mechanism and numerical methods to optimize some properties of the synthesized linkage, like the transmission angle or dynamic behavior [22].

An example that is perhaps more ubiquitous is the window casement mechanism, which allows users to open a window by a controlled amount, allowing the cleaning of both sides of the window from inside the house. Providing enough torque to operate larger double-pane windows is a challenge with a short input lever. An example of a hypocycloidal casement window operator mechanism is shown in Figure 4, and a dual-arm version that performs the same task is seen in Figure 5. Window casement mechanisms have many unique requirements—they cannot be too easy to actuate, or the window will swing in the wind, but they also need to be practical for a user to rotate them in and out. The windows are also quite heavy, so the mechanism must provide the user with a high mechanical advantage. Geared linkage designs are highly effective at addressing these and other constraints in the problem. In Figure 4, due to the gear set, the force is delivered near the end of the window during most of the motion, thus providing a large torque arm about the slider and, therefore, a high mechanical advantage for closing the window. In Figure 5, a gear set is used to split the input torque such that a portion creates a horizontal force on the slider (which sustains the weight of the window) and provides a force towards the end of the window. The result is that there is a unique push–pull moment on the window with a single input.

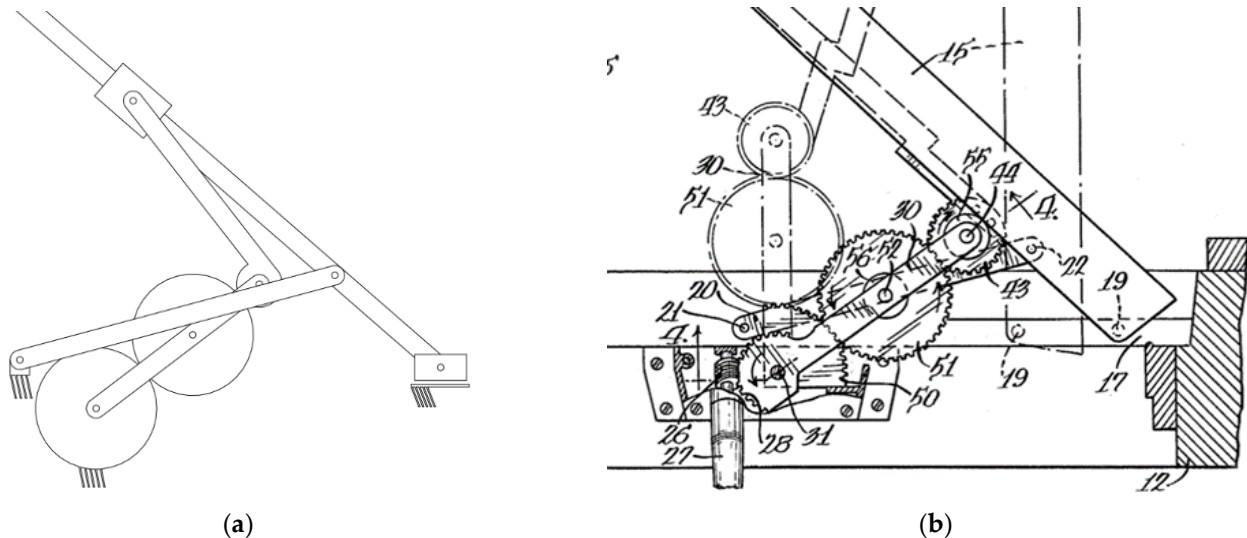

(**a**)  (**b**)

**Figure 4.** A hypocycloidal crank casement mechanism designed to securely swing a window open when the user turns the input crank, labeled 27 (US Patent #4,266,371). (**a**) A simplified view of the mechanism depicted in a half-open position. (**b**) A figure from the patent depicting the casement mechanism in a half-open and open position [23].

These previous examples assume a circular gear profile, but non-circular gears have been used creatively, and some impressively complex motions can be achieved with these gears [6,24]. The methods are quite clever, taking advantage of the inherent two degrees of freedom of a planar five-bar to precisely control two inputs at once using the non-circular profile of the gears. With that said, non-circular gears tend to be far less practical to manufacture, requiring a special custom order that is substantially more costly and can

only be operated at low speeds due to potential vibration issues. Sun et al. demonstrated a procedure for using graph theory to identify non-circular gear profiles to synthesize a GPLM. Their method was applied to the design of a transplanting mechanism, the results of which are seen in Figure 6 [25]. Transplanting in agriculture is a delicate process, as young plants tend to be quite fragile, but if performed properly, it is quite useful, as the plants can be germinated in a protected environment before being transferred to a field. Yao and Yan introduced a new algorithmic approach for optimizing the profiles of non-circular gears [26].

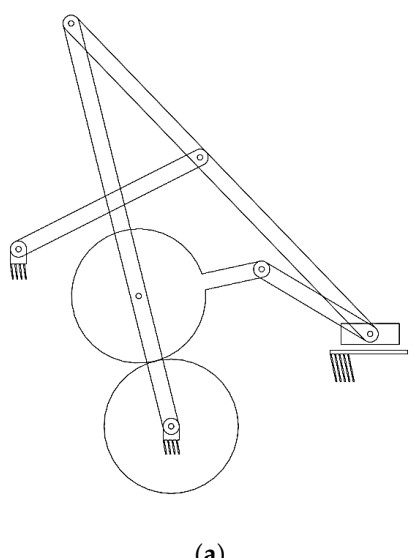

(**a**)

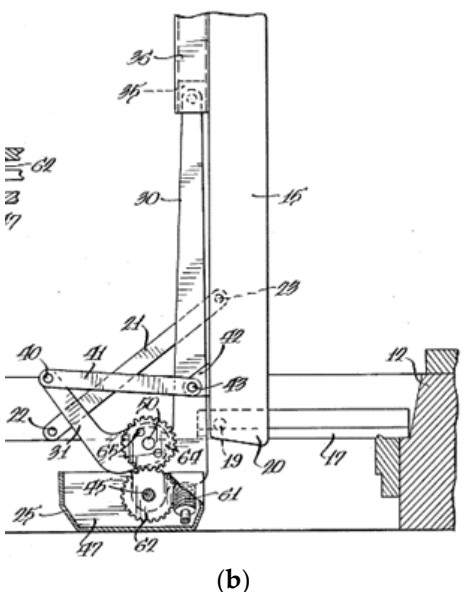

(**b**)

**Figure 5.** A dual-arm operator casement window mechanism, called dual-arm because two links act on the base of the windowsill, increases the torque (U.S. Patent #4,241,541) (**a**) A simplified view of the mechanism depicted in a half-open position. (**b**) A figure from the patent depicting the casement mechanism in the open position [27].

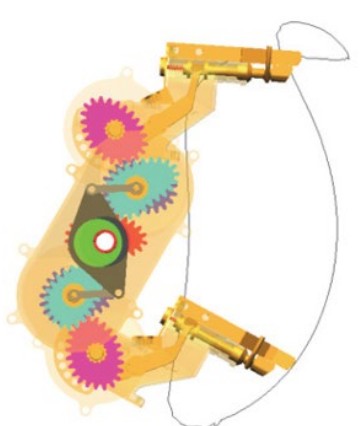

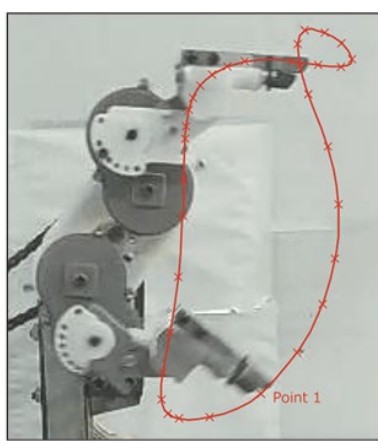

**Figure 6.** A transplanting mechanism designed with non-circular gears [25].

Volkan Parlaktaş, Engin Tanık, and Eres Söylemez investigated geared four-bar mechanisms. They maintained a single degree of freedom for the system by not affixing one of the gears to any link, allowing it to rotate freely [28]. Similarly, Sandor and Wilt studied the behavior of four-bar mechanisms in which one of the joints is an $f_2$ joint, including geared four bars [29]. $F_2$ joints are any joints that only remove a single degree of freedom from the system, like gears, cams, or pin-in-slot joints. This is opposed to $f_1$ joints, which remove two degrees of freedom from the system, like pinned joints and sliders. While these

findings open some intriguing applications for the basic four-bar mechanism, the focus of this paper is instead turned towards two or more degrees of freedom mechanisms that are constrained down to a single degree of freedom through the addition of geared connections. We propose accomplishing the synthesis task for planar GPLMs by selecting a topology, dividing it into multiple interdependent chains, and synthesizing each chain separately.

Wei-Qing Cao and Tuan-Jie Li discuss the idea of breaking a geared linkage mechanism into its independent loops, but they primarily focus on the kinematic analysis of an existing mechanism and are less concerned with designing a new mechanism for a specified task [30]. In the same vein, they also discuss the decomposition of geared mechanisms into sub-chains using tricolor graph organization [31].

Geared linkages have several potential advantages over their non-geared counterparts. First, the addition of gears allows for the creation of more complex motions and paths. Using a gear set to control the rotation or position (in the case of sliders) of one link relative to another, the relative motion can be magnified in a controlled manner. This feature is utilized by designers to create complex driven motions even further away from a grounded link. Similarly, while geared linkages offer increased complexity, they also offer precise control to designers, as the gear ratio is a design variable that designers can use to set the relative rotation rate of one link relative to another. For example, consider the mechanism shown in Figure 7. This is an example of a dwell mechanism shown in two positions, where the dwell behavior is created by controlling the gear ratio between links one and three, as well as the length of link BC. The full tracer point curve of point B is a hypocycloid, and in this case, the curve has a special property. With the input being the rotation of link 2, as the mechanism moves from the first to the second pictured position, point B on link 4 traverses a circular arc of radius R with respect to the fixed frame about point C. As a result, pivot point C does not move until the geared link moves past B* because link 4 purely rotates about C, not translating. Furthermore, link 5 (which rotates on the ground at Co) does not move—i.e., it 'dwells' during this portion of the movement of link 2.

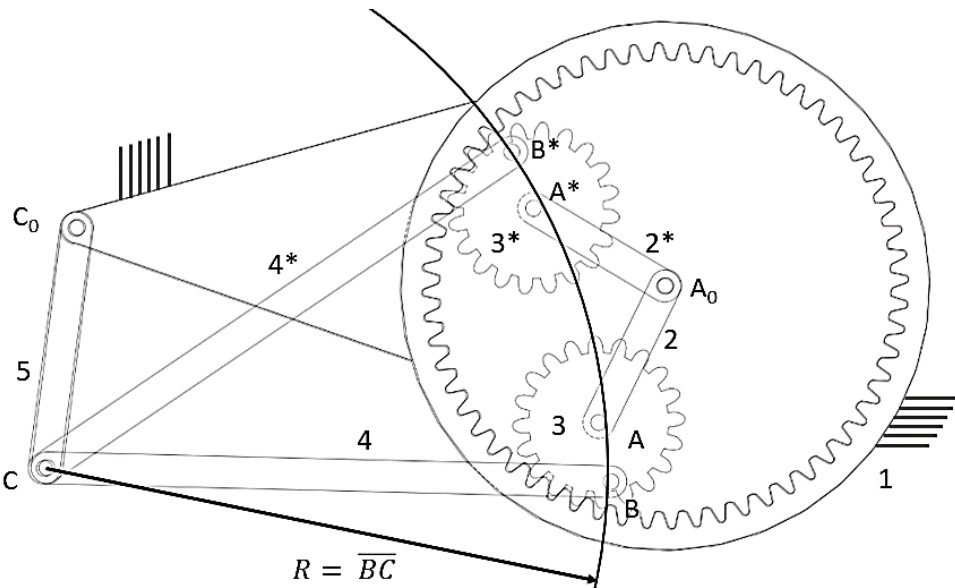

**Figure 7.** An example of a special case GPLM, where the gear ratio and link lengths are precisely controlled to create a dwell position at point C while the mechanism rotates from point B to point B*.

Another advantage of geared linkages is their improved torque transmission capabilities. Gears are naturally efficient at transmitting torque and power, but combining them with linkages allows for more interesting output motions. In comparison to conventional four bars, geared linkages can apply forces at longer distances away from the grounded link (as illustrated in the casement window mechanism in Figures 4 and 5) with less negative

impact on the mechanical advantage. As another example, a standard four-bar linkage can be converted to a cognate-geared five-bar linkage to take advantage of some of these properties of geared linkages using the topology shown in Figure 8. Here, the path tracing precision point PP1 receives torque input from both sides of the mechanism, rather than just one side that the original four bar would provide. The procedure for performing this conversion can be found in reference [9] (p. 211).

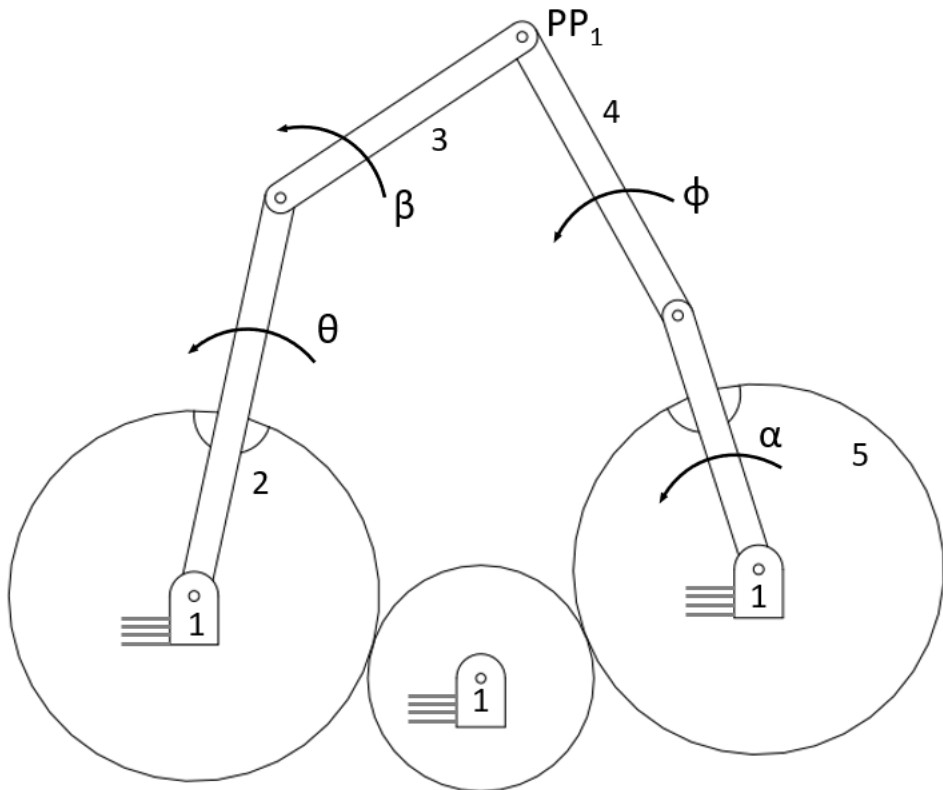

**Figure 8.** A five-bar-geared mechanism utilizing an idler to drive the input links in opposite directions. Sandor et al. show how this topology can be used to reproduce the motion of a four-bar mechanism with distinct driving properties [9] (p. 211). This mechanism could also be represented as a quadriad function generator.

While geared mechanisms have many advantages, there are some shortcomings that are valuable to mention. They can be more costly to produce than a mechanism purely comprising rigid links, especially if the required gears are not standard, with a catalog available size. Adding a gear creates a new layer of tolerancing challenges as well. If the gear teeth do not mesh tightly, the gears can introduce motion defects, but if the teeth mesh too tightly, they may inhibit the motion of the mechanism. Purchasing gears with tighter tolerances also drives up manufacturing costs. With these points considered, most designers want to first investigate topologies that do not incorporate gear elements. Even so, the benefits of implementing gears into planar linkage mechanisms in many cases outweigh the cons.

Here, we propose a problem definition and solution procedure for synthesizing geared linkage mechanisms. This method involves dividing a multiloop planar mechanism into a series of dependent chains, and then solving for the dimensions and rotation angles of these chains by considering them as complex valued vectors. The chains are referred as 'dependent,' because the values identified for one chain inform the synthesis of the next one. This approach is broadly applicable across a variety of geared mechanism topologies, is unified with other loop-based synthesis procedures, and allows designers to find a variety of candidate solutions to their problem quickly and efficiently.

## 3. Geared Linkage Synthesis Method

Here, a synthesis strategy for designing planar-geared mechanisms is suggested that can be uniformly applied across many different topologies. A feature of this method is to view the geared planar linkage mechanism as a combination of multiple open-loop-dependent dyad and triad chains. References [9,32–37] have demonstrated loop-based synthesis procedures for planar linkage mechanisms, so those core tenets are not reiterated here. Rather, the focus of this paper is on the modifications required to adapt these methods to accommodate GPLMs. The key difference in geared linkages that sets these problems apart from other loop-based synthesis problems is, of course, the inclusion of gears within the mechanism. As a result, the standard form equations must be modified to accommodate the interdependence caused by gears. The present method is a bit less flexible than some of the numerical methods mentioned in the literature review, as this approach synthesizes an exact solution for a particular problem definition rather than an approximation. However, because the basis of this method is founded in linear algebra, the computational complexity and, consequently, the computational load on any solver system that the designer uses may be relatively low. This makes the method useful as the foundational solution method behind a geared linkage synthesis program.

A designer may use the following equation to determine how the inclusion of a gear pair affects the relative rotation of the links [9] (p. 234). Here, the generic angle $\varphi$ refers to the rotation of some links with index k. The terms k + 1 and k − 1 are indexes that correspond to the links connected on either side of element k. Finally, the terms $T_{k-1}$ and $T_{k+1}$ refer to the number of teeth on the gears at the indexes k + 1 and k − 1. Index j indicates at which of the prescribed design positions this angle's calculation takes place.

$$\varphi_{j(k+1)} = \varphi_{jk} + \left[\varphi_{jk} - \varphi_{j(k-1)}\right]\frac{T_{k-1}}{T_{k+1}} \tag{1}$$

Consider again the mechanism shown in Figure 2. In this case, the gear ratio is designed as a 2:1 relationship between the 2nd and 3rd links. Practically, though, this relationship is easiest to implement as a 1:1 relationship between the 3rd link and a gear affixed to the ground, as shown in the dyad chain in Figure 9.

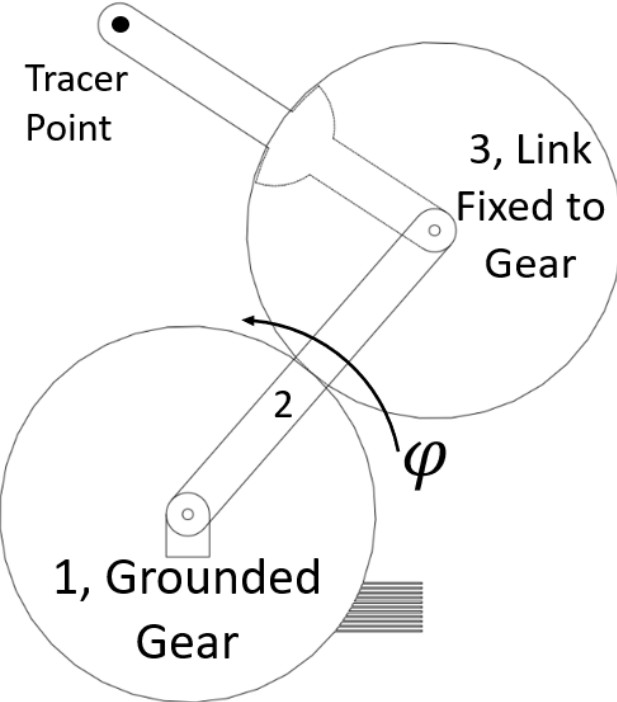

**Figure 9.** A dyad chain shown with an added gear pair.

Equation (2) shows the standard form equations for a triad expressed in a matrix form. In these equations, **W**, **Z**, and **V** represent (in complex number vector form) the first, second, and third links in the triad chain. $\beta$ represents the angular displacement of **W**, $\alpha$ represents the angular displacement of **Z**, and $\gamma$ represents the angular displacement of **V**. Each of these angular displacements is measured from one prescribed position to the next. As a result, with four prescribed positions, each angular displacement variable has the following three values: the displacements from the first position to the second, third, and fourth. In Equation (2), these angles are indexed to indicate the position they correspond to using subscripts 2–4. Finally, $\delta_j$ is the positional displacement vector between position one and the $j^{\text{th}}$ position. Note that any quantities expressed in bold font are vector quantities, while non-bolded quantities are scalars. Using Equation (1), any of the angular displacements can be related to each other—the designer chooses which links to relate by choosing where the gear pair is positioned in the kinematic chain. Note that Equation (2) assumes all members are rigid bodies and do not yield or bend under loads. In most cases, this is a good approximation, but for mechanisms that deal with high loads or are built with flexible members (including a special class of machine known as compliant mechanisms), additional dynamic body calculations are required. While this is a vast field of research, a few examples can be found in references [38–43].

$$\begin{bmatrix} e^{i\beta_2} - 1 & e^{i\alpha_2} - 1 & e^{i\gamma_2} - 1 \\ e^{i\beta_3} - 1 & e^{i\alpha_3} - 1 & e^{i\gamma_3} - 1 \\ e^{i\beta_4} - 1 & e^{i\alpha_4} - 1 & e^{i\gamma_4} - 1 \end{bmatrix} \begin{bmatrix} W \\ Z \\ V \end{bmatrix} = \begin{bmatrix} \delta_2 \\ \delta_3 \\ \delta_4 \end{bmatrix} \tag{2}$$

The standard form shown in Equation (2) may be modified to accommodate a dyad or quadriad according to the relationships shown in Figure 10. It can also be noted that each additional gear connection between two links decreases the degrees of freedom by one, according to Gruebler's equation [9]. The designer must take this into consideration when selecting the number of prescribed positions for the problem at hand. Thus, Table 1 is helpful in guiding this decision, describing a triad containing one gear constraint and a single degree of freedom. Although the open-loop chains shown in Figures 11 and 12 have more than a single degree of freedom on their own, they may be reduced to a single degree of freedom by combining the chains with other links to form closed-loop mechanisms.

**Table 1.** Geared triad motion generator free choices.

| Number of Prescribed Poses | Scalar Equations | Scalar Unknowns | Free Choices | Solutions |
|---|---|---|---|---|
| 2 | 2 | 7 (**W, Z, V**, $\alpha_2$) | 5 | $\infty^5$ |
| 3 | 4 | 8 (above + $\alpha_3$) | 4 | $\infty^4$ |
| 4 | 6 | 9 (above + $\alpha_4$) | 3 | $\infty^3$ |
| 5 | 8 | 10 (above + $\alpha_5$) | 2 | $\infty^2$ |
| 6 | 10 | 11 above + $\alpha_6$) | 1 | $\infty$ |
| 7 | 12 | 12 (above + $\alpha_7$) | 0 | Finite |

This table is for a geared triad with prescribed motion. The rotation of one of the gears is known (free choice or prescribed), and therefore, the rotation of the other link rigidly connected to a gear is known. For a geared triad, used for path generation with prescribed timings in five positions, a designer may instead use their free choices on one of the vectors **W**, **Z**, or **V**. Table 1 is modified to show the properties of quadriad chains in Table 2. The findings of Table 2 apply to geared quadriad chains, which have three links rigidly constrained to gears, including one free choice and a final angle prescribed by the problem. In this context, free choices may be any angle value that is not prescribed in the problem. As identified in Tables 1 and 2, for problem definitions that do not have

much-prescribed data, designers need to set the value of some angles so that Equation (2) can be evaluated directly. For geared mechanism synthesis, choosing a gear ratio between two links eliminates several of a designer's freely chosen variable values, as the second link must rotate relative to the first according to the gear ratio between them.

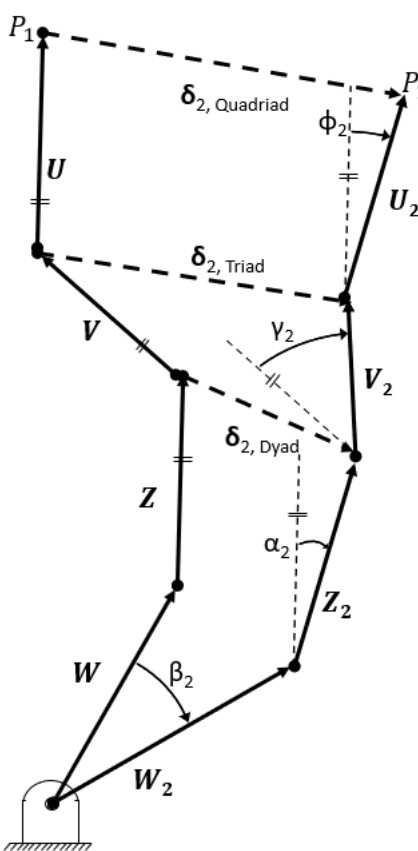

**Figure 10.** A vector forms the depiction of a quadriad chain with intermediate chains labeled. A triad chain does not include the **U** vector, while a dyad chain does include either the **U** or **V** vectors. The quadriad chain is depicted in an initial position and a second position resulting from the rotations $\beta_2$, $\alpha_2$, $\gamma_2$, and $\phi_2$. Note that the vector $\delta_{\text{triad}}$ in this figure is equivalent to the vector $\delta_2$ in Equation (2). Figure 10 reveals how Equation (2) might be reformulated for quadriad or dyad chains, as shown in Appendix C.

**Table 2.** Geared quadriad motion generator * free choices.

| Number of Prescribed Poses | Scalar Equations | Scalar Unknowns | Free Choices | Solutions |
|:---:|:---:|:---:|:---:|:---:|
| 2 | 2 | 9 (**W, Z, V, U**, $\alpha_2$) | 7 | $\infty^7$ |
| 3 | 4 | 10 (above + $\alpha_3$) | 6 | $\infty^6$ |
| 4 | 6 | 11 (above + $\alpha_4$) | 5 | $\infty^5$ |
| 5 | 8 | 12 (above + $\alpha_5$) | 4 | $\infty^4$ |
| 6 | 10 | 13 above + $\alpha_6$) | 3 | $\infty^3$ |
| 7 | 12 | 14 (above + $\alpha_7$) | 2 | $\infty^2$ |
| 8 | 14 | 15 (above + $\alpha_8$) | 1 | $\infty$ |
| 9 | 16 | 16 (above + $\alpha_9$) | 0 | Finite |

* These same relations are applicable for a four-bar function generator [9] (p. 102).

- **Geared Triad Chains:**

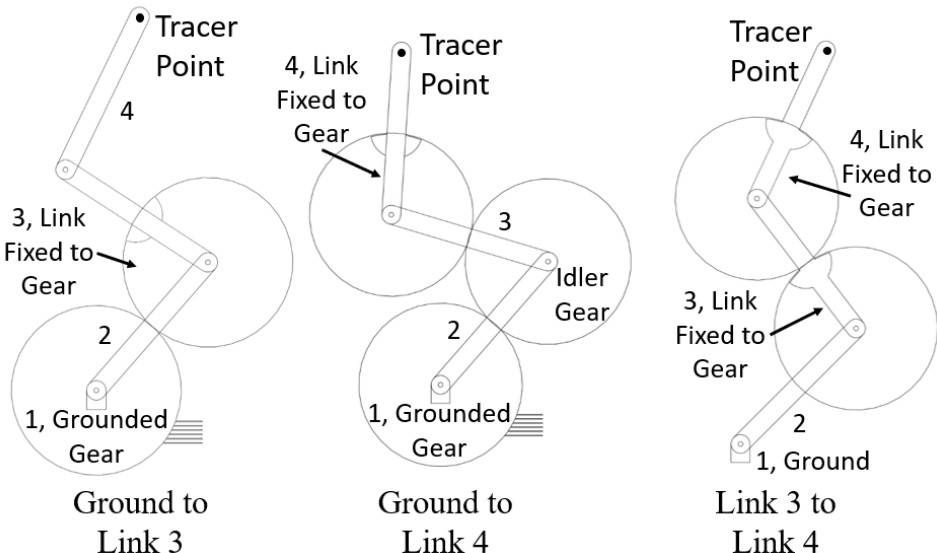

**Figure 11.** The three primary cases of a geared triad linkage chain. On the **left**, the rotations of links 1 and 3 are related. In the **middle**, the rotations of links 1 and 4 are related, while on the **right**, the gears relate to links 3 and 4.

- **Geared Quadriad Chains:**

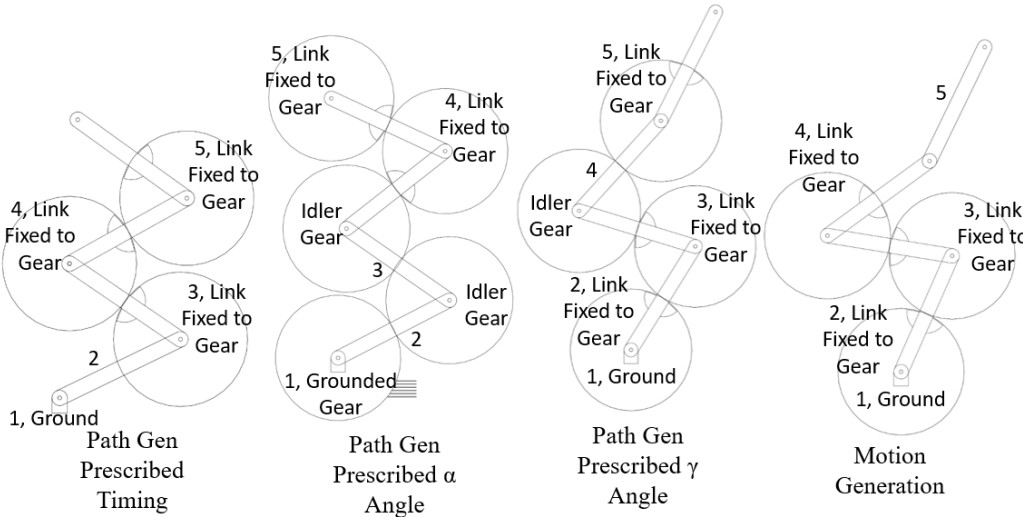

**Figure 12.** Example cases of the geared quadriad chains. This list is not comprehensive, and many additional variations exist, especially when considering chains with additional degrees of freedom.

The classic rendering of gears almost always involves two spur gears rotating opposite to one another, and while this form is effective in many circumstances, designers should also not neglect the potential of geared linear translation elements. The present method is amenable to synthesizing geared topologies that incorporate these elements, like the geared five-bar shown in Figure 13, which incorporates a rack and pinion; in fact, linear translation elements may be identified in the results of Equation (2) without any prior intention on the designer's part to include them. If one of the vector links (**W**, **Z**, **V**) found from this equation has a magnitude that is several orders of magnitude larger than the other links, it likely indicates that a linear-translating or sliding element may function better in that link's place. An exceptionally long link typically indicates that, relative to the scale of the rest of the mechanism, the motion of the distal end of that long link is approximately a

straight line with minimal rotation. Thus, rather than implementing the absurd solution of an infinitely long, rigid link, designers may replace the long link with a linear sliding element. A brief numerical example of a geared sliding/linear translation element is shown in Appendix A. An example of a relatively long link being approximated as a linear slider is shown in Appendix B.

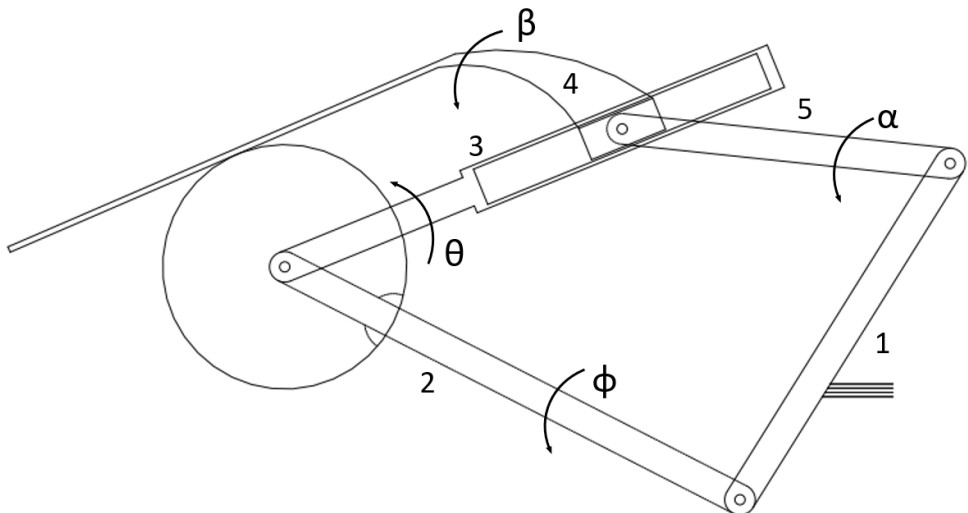

**Figure 13.** Geared mechanism topologies are also readily adapted to accept sliding components. One example is shown above, which uses a rack and pinion pair to drive the slider link [44] (p. 43).

### 4. Geared Planar Five-Bar Synthesis Example

To demonstrate this procedure, a synthesis example is provided using the prescribed data shown in Table 3.

**Table 3.** Prescribed data.

| Prescribed Position | Moving Plane ($\gamma$) Angle | $\delta$ |
|---|---|---|
| 1 | - | 0 |
| 2 | 10 | 2 + 2i |
| 3 | 50 | 4 + 5i |
| 4 | 75 | 7 + 4i |

Using the presented method, for GPLM design problems, the gear ratio is the free choice of the designer, along with which links are geared together. For this example, the second link in the triad chain is connected via a gear pair to the ground, and the gears are used to implement a ratio of two to one between links two and three (As in Figure 11, "Ground to Link 2"). To create a one-degree-of-freedom closed-loop mechanism from this triad chain, a non-geared dyad chain was added. The overall topology was then a five-bar GPLM with a gear pair between links one and three.

Equation (1) (gear tooth to ratio) is used to find the gear ratio between links one and three, which produces the desired 2:1 ratio between links two and three. The equation is rewritten as Equation (3), using 2 and 1 as arbitrary angles with a ratio of 2:1 for links three and two, respectively.

$$2 = 1 + [1 - 0]\frac{T_1}{T_3} \tag{3}$$

Solving Equation (3) for the teeth ratio between the grounded gear and link three yields a ratio of one. The standard form equations shown in Equation (2) are rewritten to account for the geared connection between links one and three, as seen in Equation (4).

$$
\begin{bmatrix}
e^{i\beta_2} - 1 & e^{i\beta_2*2} - 1 & e^{i\gamma_2} - 1 \\
e^{i\beta_3} - 1 & e^{i\beta_3*2} - 1 & e^{i\gamma_3} - 1 \\
e^{i\beta_4} - 1 & e^{i\beta_4*2} - 1 & e^{i\gamma_4} - 1
\end{bmatrix}
\begin{bmatrix} W \\ Z \\ V \end{bmatrix}
=
\begin{bmatrix} \delta_2 \\ \delta_3 \\ \delta_4 \end{bmatrix}
\tag{4}
$$

For this most basic case of a triad with four prescribed motion positions, the angles $\beta_{2-4}$ are free choices, meaning that the designer may choose the value of each of these variables, while the angles $\gamma_{2-4}$ are prescribed in the problem. Because the gear ratio is taken as a free choice, in this example, the $\alpha_{2-4}$ angles are also known, simplifying to $\beta_{2-4}$*r, where r = 2. Equation (4) is directly solvable using linear algebra to identify values for the vectors **W**, **Z**, and **V**. Written out explicitly in units of degrees for the given problem, Equation (4) becomes Equation (5).

$$
\begin{bmatrix}
e^{-\frac{\pi}{4}i} - 1 & e^{-\frac{\pi}{2}i} - 1 & e^{0.174i} - 1 \\
e^{-1.309i} - 1 & e^{-2.618i} - 1 & e^{0.873i} - 1 \\
e^{-1.658i} - 1 & e^{-3.316i} - 1 & e^{1.309i} - 1
\end{bmatrix}
\begin{bmatrix} W \\ Z \\ V \end{bmatrix}
=
\begin{bmatrix} 2 + 2i \\ 4 + 5i \\ 7 + 4i \end{bmatrix}
\tag{5}
$$

This matrix may be evaluated directly to solve **W**, **Z**, and **V**.

To make the overall mechanism a closed loop with one degree of freedom, a non-geared dyad pair was added. This chain has the same prescribed motion as the triad chain ($\delta_{2-4}$, $\gamma_{2-4}$), but the standard form equations cannot be evaluated directly through linear algebra because there are too many unknowns.

$$
\begin{bmatrix}
e^{i\beta_2} - 1 & e^{i\alpha_2} - 1 \\
e^{i\beta_3} - 1 & e^{i\alpha_3} - 1 \\
e^{i\beta_4} - 1 & e^{i\alpha_4} - 1
\end{bmatrix}
\begin{bmatrix} W \\ Z \end{bmatrix}
=
\begin{bmatrix} 2 + 2i \\ 4 + 5i \\ 7 + 4i \end{bmatrix}
\tag{6}
$$

While the triad in four positions has three vector equations and three unknowns, the dyad in four positions has two vector equations and six unknowns (vectors **W**, **Z**, and angles $\beta_3$ and $\beta_4$). As a result, here, the compatibility linkage technique is employed to determine the dimensions of the dyad. The compatibility linkage is a mathematical technique that makes it possible to find exact solutions to the standard form equations despite having more unknowns than equations and is covered extensively in the existing literature [9,32,45–47]. Using this technique, the designer has one free choice available to them, which is selected as the angle $\beta_2$ and set as 205.35°.

With all these parameters set, the complete mechanism is synthesized, as seen in Figure 14, and the numerical values of the vectors in their first position are shown in Table 4. The mechanism is depicted in its first and fourth positions in Figure 15, with the third link in the triad chain and the second link in the dyad chain fused into a single link. This is possible because these two links have the same prescribed rotations; fusing them into a single link reduces the degrees of freedom of the overall system by one while maintaining their target rotations, though it is possible that this step could introduce motion defects. In this case, the mechanism worked as intended, reaching the four positions in an arching motion. The coupling link was extended out to achieve a larger sweeping arc motion, which could traverse a greater distance in a single cycle. Assuming it was desired to operate this mechanism with a motor driving in one direction, an additional dyad was added to the base mechanism synthesized in this example to turn it from a rocker–rocker to a crank–rocker-type mechanism. The addition of this driving dyad also eliminates the risk of a branch defect occurring at the fringes of the mechanism's range of motion. By driving the first link in this driving dyad, which has full rotational freedom (link 6 of Figure 2), the resulting mechanism can be driven by a motor rotating in one direction and is fully defect-free. A depiction of this form is shown in Figure 2.

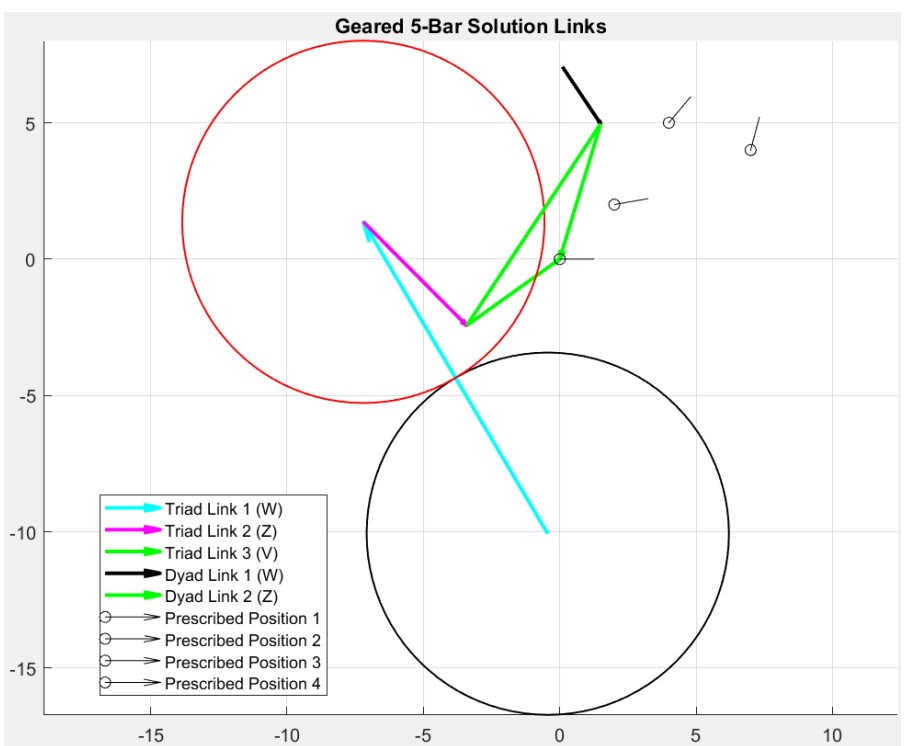

**Figure 14.** The solution mechanism shown in its first position. Here, the first, second, and third links in the triad correspond to the vectors **W, Z,** and **V**, respectively. Similarly, the first and second links in the dyad correspond to the vectors **W** and **Z**. The proximal ends of the first link of the triad and the first link of the dyad are connected to the ground.

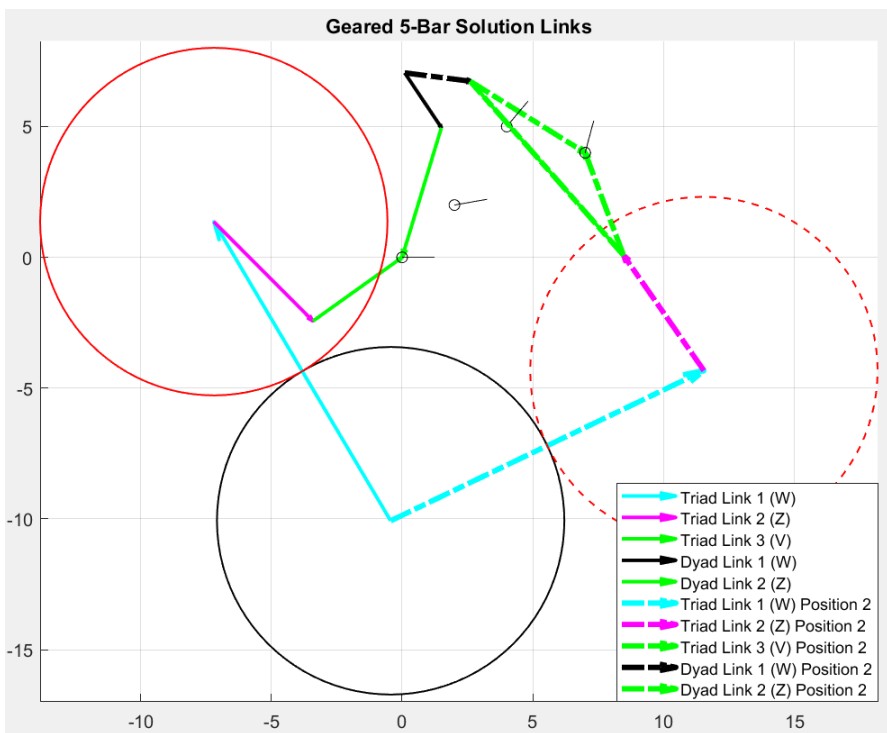

**Figure 15.** The solution mechanism is shown moving from the first (solid line) to the fourth (dashed line) prescribed position. In this figure, a ternary link was formed from the links 'dyad link 2' and 'triad link 3' shown in Figure 14, which was possible because those links had the same prescribed rotations.

**Table 4.** Calculated link length and angle data.

| Component | Value |
|:---:|:---:|
| $\mathbf{W_{Triad}}$ | $-6.7635 + 11.4357i$ |
| $\mathbf{Z_{Triad}}$ | $3.7905 - 3.8019i$ |
| $\mathbf{V_{Triad}}$ | $3.4121 + 2.436i$ |
| $\mathbf{W_{Dyad}}$ | $1.4042 - 2.0949i$ |
| $\mathbf{Z_{Dyad}}$ | $-1.5015 - 4.9586i$ |
| $\beta_{3,\mathbf{Dyad}}$ | $121.0779°$ |
| $\beta_{4,\mathbf{Dyad}}$ | $48.8814°$ |

Using the software "Lincages," two candidate four bars were designed using the same set of prescribed positions and angles to compare against this geared five-bar mechanism. The links and transmission angles for each of these alternate mechanisms are shown in Figures 16 and 17. The transmission angles of the geared five bar over its range of motion are shown in Figure 18. The transmission angle of the geared five bar dips down to 34° at its minimum, and the average is around 63°, which is well within acceptable levels. This is also noticeably better than the two primary four-bar candidates, which both fall as low as 5° at different points in their motion. Pivot locations are also valuable to consider. In this respect, the mechanism shown in Figure 16 is clearly less desirable as it has a long driving link. The lengths of the largest link in Figure 17 and the geared five-bar are more similar (the four-bar's longest link is approximately 1.45 times longer), but the pivot locations are improved in the geared five-bar mechanism. The configuration is much tighter because the coupling link is kept shorter. The geared mechanism would likely end up being a bit more expensive to manufacture, but in this example, the addition of a pair of gears yielded several advantages over four-bar mechanisms designed with an equivalent set of prescribed positions. The average transmission angle was higher, and the minimum transmission angle also never dipped below 30°, which is a value commonly recommended as a minimum to maintain proper motion. Additionally, the design maintains a tight profile, fitting into a tighter space than the four-bar designs.

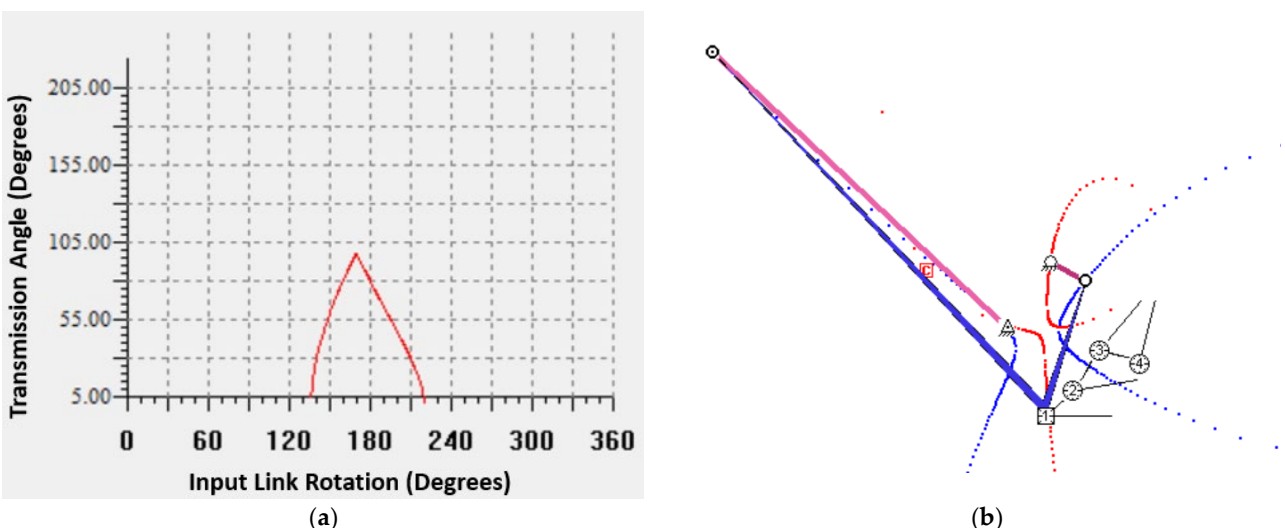

(**a**)          (**b**)

**Figure 16.** A candidate four-bar linkage designed with the same set of prescribed positions. (**a**) The transmission angle data of the four-bar linkage throughout its range of motion. (**b**) A visualization of the synthesized four-bar linkage, including the prescribed positions and Burmester curves.

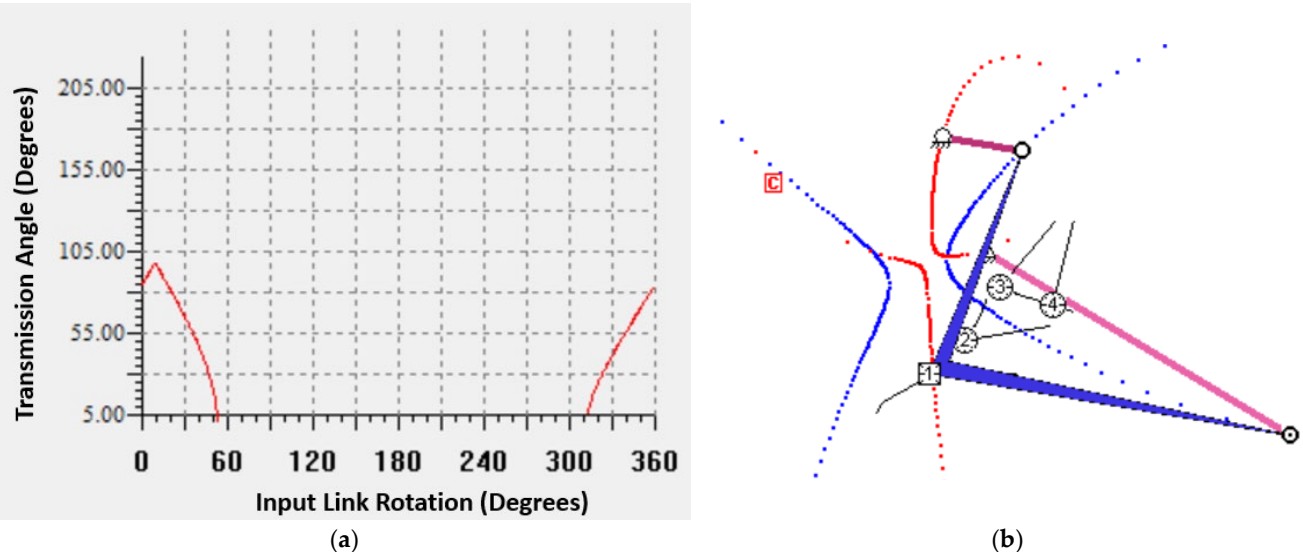

(**a**)                                                                       (**b**)

**Figure 17.** A second candidate four-bar linkage. (**a**) The transmission angle data of the four-bar linkage throughout its range of motion. (**b**) A visualization of the synthesized four-bar linkage, including the prescribed positions and Burmester curves.

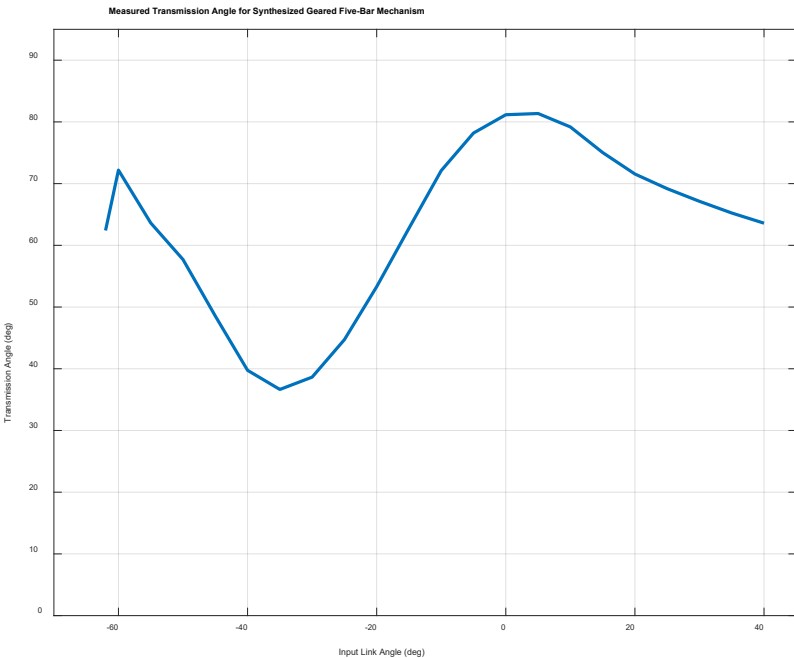

**Figure 18.** The transmission angle data throughout the full range of motion of the synthesized geared five-bar mechanism.

## 5. Conclusions

The present method is widely applicable in the synthesis of geared planar linkage mechanisms. At its core, this is an analytical approach, and the formulas are readily adapted to accommodate a variety of geared topologies. Furthermore, because of the foundational underpinnings in linear algebra, the present method makes an effective general synthesis tool in a numerical implementation and can reduce the computational load relative to other numerical methods. A hybrid approach could also be quite effective, with a mechanism initially being synthesized analytically and then optimized through a genetic algorithm or other numerical approach. The provided numerical example demonstrates the effectiveness of the loop-based synthesis approach for geared planar linkage mechanisms as a

robust solution procedure, and it paired naturally and intuitively with existing loop-based synthesis techniques.

While presently effective, there remain several opportunities for future work in this area. To segment a candidate mechanism topology into discrete loops, a designer must first decide on a topology that is fit for the task at hand, though building an intuition for which topologies are effective at addressing any given problem is no easy task. Therefore, identifying and cataloging the properties of practical geared topologies or making a systematic methodology for optimizing topology selection would be valuable additions to this work. Analytical loop-based synthesis methods like the present approach rely on the designer making some number of 'free choices'—design variables like rotation angles which are not specified in this problem. Drastically different solutions may result depending on the values selected for these free choices, with some resultant mechanisms faring much better than others with respect to common benchmarks like the transmission angle or mechanical advantage. As a result, there is another opportunity for future work in identifying strategies and applicable tactics to optimize free-choice selection. More work could also be performed in formulating a consistent procedure for rejecting free-choice values that yield solutions with motion defects. Motion defects, like branch or circuit defects, can occur when a mechanism must pass through a toggle position or be disassembled to reach the next prescribed position. These types of motion defects are some of the most common causes of a candidate mechanism failing, so rooting them out consistently is a valuable addition to the method. It should be possible to expand this method to accommodate three-dimensional topologies in addition to planar ones, like spherical geared linkages; this investigation is also left to future work.

**Author Contributions:** Conceptualization, S.M. and A.E.; methodology, S.M. and A.E.; software, S.M.; validation, A.E.; formal analysis, S.M. and A.E.; investigation, S.M. and A.E.; resources, A.E.; writing—original draft preparation, S.M.; writing—review and editing, A.E.; visualization, S.M. and A.E.; supervision, A.E. All authors have read and agreed to the published version of the manuscript.

**Funding:** This research received no external funding.

**Data Availability Statement:** Data are contained within the article.

**Conflicts of Interest:** The authors declare no conflicts of interest.

## Appendix A. Rack and Pinion Gears Using a Triad Chain

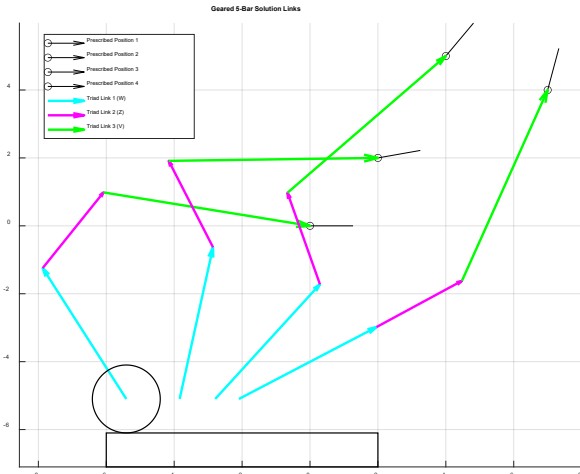

**Figure A1.** A graphical example of a sliding element (or linear translation element) introduced into a geared chain. Here, the gear relationship is formed between the angle of the link **W** in the triad chain and ground. In this case, the effect is that link **W,** which is connected to the pinion gear, is rolling with respect to the rack which is connected to ground.

To introduce this sliding element, the standard form equations (Equation (2)) are modified as shown in Equation (A1).

$$\begin{bmatrix} e^{i\beta_2} - 1 & e^{i\alpha_2} - 1 & e^{i\gamma_2} - 1 \\ e^{i\beta_3} - 1 & e^{i\alpha_3} - 1 & e^{i\gamma_3} - 1 \\ e^{i\beta_4} - 1 & e^{i\alpha_4} - 1 & e^{i\gamma_4} - 1 \end{bmatrix} \begin{bmatrix} W \\ Z \\ V \end{bmatrix} = \begin{bmatrix} \delta_2 + r\beta_2 \\ \delta_3 + r\beta_3 \\ \delta_4 + r\beta_4 \end{bmatrix} \tag{A1}$$

In this equation, an additional term was added to the right-hand side of the equation which accounted for the movement of the proximal end of the triad vector chain. The value 'r' is a scalar value set by the designer that relates the rotation of link 2 ($\beta_j$) to the displacement of its proximal end. Setting the equations up in this way forces a linear translation element at the proximal end of the triad chain, but it is also possible to identify sliding/linear translating elements without preselecting the sliding element's position.

### Appendix B. Synthesizing a Slider Using a Dyad Chain

Consider the dyad shown in Figure A2. The link **W**, or the first link in the dyad chain, is several orders of magnitude longer than the second link **Z**. As a result, a zoomed-in view of the distal end of link **W** appears to trace a straight line. Rather than creating a link that is over 6000 units long, the entire link **W** can be replaced by a sliding element connecting ground to link **Z**. The path of this sliding element should be a straight line perpendicular to the ground pivot of the original, long version of link **W**.

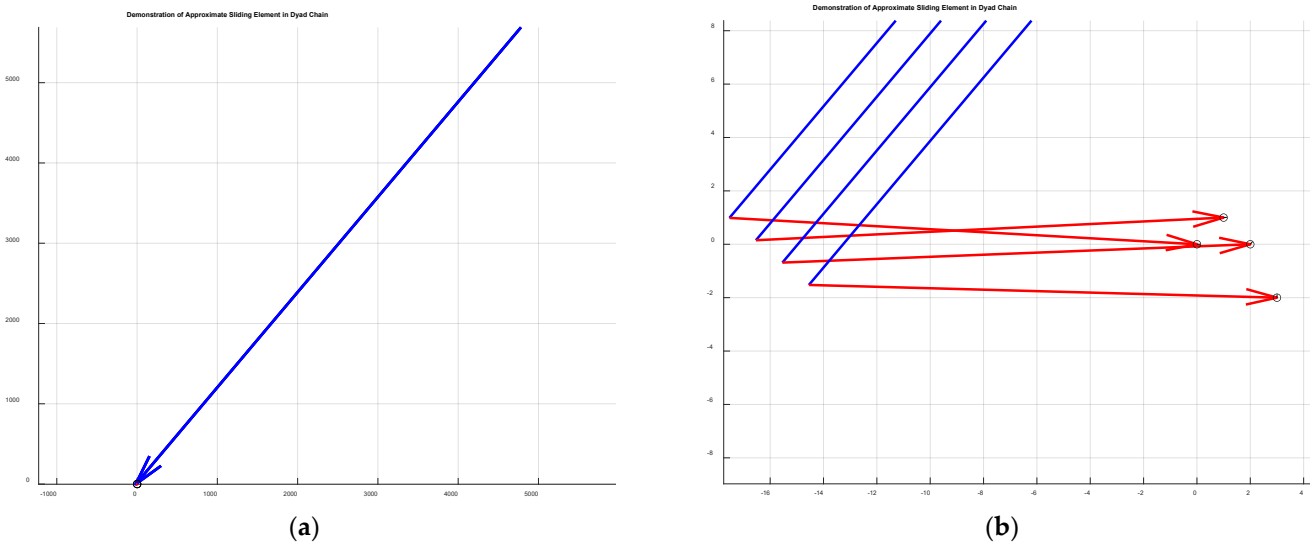

| (a) | (b) |

**Figure A2.** (**a**) A dyad chain with a disproportionately long link **W**, such that the link **Z** can hardly be seen at its distal end. (**b**) A close-up view of the distal end of link **W**, showing the connections formed by link **Z** between the end of link **W** and the prescribed positions. Notice that, at this scale, the distal end of link **W** approximately traces a straight line.

### Appendix C. Standard Form Equations of Dyad and Quadriad Chains

Standard form equations of a dyad:

$$\begin{bmatrix} e^{i\beta_2} - 1 & e^{i\alpha_2} - 1 \\ e^{i\beta_3} - 1 & e^{i\alpha_3} - 1 \end{bmatrix} \begin{bmatrix} W \\ Z \end{bmatrix} = \begin{bmatrix} \delta_2 \\ \delta_3 \end{bmatrix} \tag{A2}$$

Standard form equations of a quadriad:

$$\begin{bmatrix} e^{i\beta_2} - 1 & e^{i\alpha_2} - 1 & e^{i\gamma_2} - 1 & e^{i\phi_2} - 1 \\ e^{i\beta_3} - 1 & e^{i\alpha_3} - 1 & e^{i\gamma_3} - 1 & e^{i\phi_3} - 1 \\ e^{i\beta_4} - 1 & e^{i\alpha_4} - 1 & e^{i\gamma_4} - 1 & e^{i\phi_4} - 1 \\ e^{i\beta_5} - 1 & e^{i\alpha_5} - 1 & e^{i\gamma_5} - 1 & e^{i\phi_5} - 1 \end{bmatrix} \begin{bmatrix} W \\ Z \\ V \\ U \end{bmatrix} = \begin{bmatrix} \delta_2 \\ \delta_3 \\ \delta_4 \\ \delta_5 \end{bmatrix} \tag{A3}$$

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
