# Peer review of "Synthesis of Geared Planar Linkage Mechanisms through the Segmentation of Multiloop Mechanisms into Discrete Chains"

_machines, doi:10.3390/machines12030182_

Round 1

Reviewer 1 Report

Comments and Suggestions for Authors

Dear Authors

This is an interesting article about synthesis of Geared Planar Linkage Mechanisms. This problem is important for the development of new mechanisms, especially path generation mechanisms.

However, the article has some flaws that should be clarified.

A part of the references is quite old.

The ancient mechanism from Antikythera is the oldest known gear mechanism dating to the 2nd century BC.

In order to clarify the methodology, assumptions should be specified, including the assumption of a rigid body.

Only advantages of geared linkages have been described.

It is also important to mention the disadvantages of a gear mechanism, including the precision mainly related to the manufacturing accuracy of the teeth, and possible motion defects.

How does the computational complexity of proposed method compare to other methods?

The kinematic characteristics of proposed mechanism could be described in more detail as the 4 points path planning is not very representative.

The conclusions could have been more elaborate.

Other comments

Fig. 4 is hardly readable

Fig A2 there is to much green color and the red link is missing

Author Response

Reviewer One Response:

Thank you for your thoughtful feedback! We’ll respond to each of your listed requests for modification.

  • Old references: This is a fair observation to make, as many of the references included in the manuscript are from the 1980s. However, part of the reason these references are included is to help paint a picture of the timeline of historical geared mechanism research. Specifically, the trend that the methods primarily being investigated have shifted from analytical approaches to numerical ones. We’ll try to make this point clearer in the manuscript, but we have also added quite a few additional references to try to make the list feel more thorough.
  • Thank you for raising the Antikythera mechanism! It was overlooked in our initial search. We have added a few references to the device in the manuscript.
  • Assumptions should be specified – This is a great point! We have clarified a few of the key assumptions in the text. We have taken more care to point out some assumptions in the text, including calling out the rigid body assumption where we discuss the formulation of the analytical equations.
  • Only advantages of geared mechanisms have been described – This is a good point! It is valuable to point out some of the shortcomings of our proposed approach, including the ones that you stated in your response.
  • Computational complexity – We’ve added a statement comparing the computational complexity to other methods when it is first introduced.
  • Kinematic characteristics of proposed mechanism could be described in more detail – The precise meaning of this comment is unclear to us. We are not necessarily advocating the design of any particular mechanism. We have attempted to clarify the meaning of precision position synthesis.
  • Conclusions could have been more elaborate – This comment aligns with the comments of other reviewers. We have added additional commentary to the conclusion summarizing the work and discussing further options for future work.
  • Figure 4 not readable – We have modified one-half of Figure 4. The other half is a patent figure which I am not able to edit, though the text on the figure is not critical to the reader’s understanding of the figure.
  • Figure A2 – We have reexamined figure A2, and it does appear to be correct. The large volume of green color is because the red link in Figure A1 has been replaced with a green link. Additionally, a third green line has been added in each prescribed position. This was done to indicate that the red and green links in Figure A1 are actually one solid link in the final mechanism. A statement was added to the appendix to clarify this relationship.

Thank you again for your comments! We hope that our responses here clarify any remaining doubts or concerns that you had about the manuscript!

Reviewer 2 Report

Comments and Suggestions for Authors

Author Response

Thank you for this thoughtful feedback, and for your support of the manuscript! We’ll respond to each of your comments below.

First, you mention denoting the links in each figure with Arabic numerals. We have taken care of this for Figure three depicting the planetary gear train. Unfortunately, the other two figures are not our own creations, they’ve been reproduced from other sources, so we will not be able to modify them.

We’ve added an explanation of the index j in equation 1.

The scales have been adjusted to match in Figure 2.

You make a good point about the hatching in Figure 6 looking a bit different. We’ve adjusted the hatching pattern so that they are the same. It should be just one ground location.

In Figure 7, all five bars are shown (ground is counted as link 1). We’ve added hatching to the ground points to make their purpose clearer.

Additional description has been added in the requested lines regarding the geared dwell topology.

Thank you for catching that Figure 9A was an improper citation in the text! That caption should have read Figure 10A and has now been corrected.

You make several good points about the final example that we provide. We have tried to add significantly more numerical detail to this example to make the procedure and solution clearer. It is also a fair point that Figure A2 is a bit unclear. We have tried to clarify this Figure, taking some of your suggestions into account. The figure is now only shown in 2 positions, the first and the last,  rather than 4.

Additional details have been added to the conclusion. Theoretically you could implement this method for spherical mechanisms by putting gear teeth on a spherical joint, but for the scope of this paper we are primarily concerned with planar mechanisms.

Reviewer 3 Report

Comments and Suggestions for Authors

## General description

In this manuscript, the authors discuss the design synthesis of geared planar linkages, namely, linkages that include both lower kinematic pairs (such as rotary or prismatic joints) and gears. The authors have done previous research on the synthesis of planar linkages: see Refs. [4,17-20,22,24,29] in the bibliography.

There is a vast literature on geared linkages. However, the bibliography appears limited, with only 29 works cited, out of which 8 are by at least one of the authors. A tentative list (by no means complete) of references that could be usefully cited is as follows:

1. "Extension of Freudenstein's equation to geared linkages", Rao - Sandor 1971;

2. "Optimization of parameters for specified path generation using an atlas of coupler curves of geared five-bar linkages", Zhang et al. 1984;

3. "Mobility criteria of geared five-bar linkages", Ting 1994;

4. "A nine-bar linkage for mechanical forming presses", Tso - Liang 2002;

5. "Path curvature of a geared seven-bar mechanism", Pennock - Sankaranarayanan 2003.

The manuscript is structured as follows: after the introduction in Sec. 1, Sec. 2 presents the kinematic analysis of a geared linkage. The manuscript does not include an experimental section, which can be understandable in context, but a numerical example is reported in App. A. Conclusions are then drawn in Sec. 3, and directions for future work are proposed, which are however too vague: what are the "optimal free choice values" and which kind of "defects" should be avoided (line 229)? I suggest to add a paragraph detailing the paper structure at the end of Sec. 1.

## Notes and problems

- The scope of the contribution appears limited. Aside from a brief state of the art in Sec. 1, the main part is Sec. 2, detailing an analysis method based on complex numbers: while technically correct, this method is not novel in itself (see for instance "Mechanics of Machines", Cleghorn-Dechev 2016). 

- The manuscript is in a draft version that includes comments and is quite hard to read. I suspect this draft was uploaded by mistake instead of the final version.

## Style and organization

The quality of writing is good. The introduction and the conclusions are well written. The writing technical accuracy is fair, though some points could be clarified (see Sec. "Other minor issues" in this review). The title is somewhat unclear: what is meant by "dissolution" here? 

The notation does not respect typographic conventions: subscripts in delta_2 and so on in Eq. (2) should not be bold. All symbols should be defined before being introduced: see W, Z and V on p. 7. Symbols should be used coherently: is delta_triad in Fig. 9 equivalent to delta_2 in Eq. (2)?

The quality of the illustrations is not adequate for publication, as they are very small, blurry and hard to read. In Fig. A1, it is necessary to define T1 and so on (in the legend); Fig. A2, reporting several graphical elements together, is especially hard to read, especially when printed in black and white.

## Other minor issues

- The mechanism in Fig. 4 is from a 1979 patent, so describing it as "novel" (line 70) seems excessive.

- Point B in Fig. 6 does not move along a "circular arc" (line 118), as the curve traced by B is a hypocycloid.

- With respect to what is the "angular displacement" beta measured (line 164)? The definition of delta_j as "the positional displacement vector between each of the prescribed positions" (line 166) appears vague.

- In Fig. 10, all kinematic chains have two Degrees-of-Freedom (DoFs), so the analysis in lines 174-175 ("single degree of freedom") is inaccurate. Also, notice that the chains in Fig. 11 have two DoFs, except for the second one from the right, which has only one DoF.

- Which position is considered in Tab. 1 (first column)? Is it the position of the end point P?

- The last paragraph of p. 8 should be rewritten: what is meant by "free choice" here? 

- Table 2 should not be split between separate pages; also, a table footnote (line 192) is present, but it is not clear to which part of the table it refers.

- The claim "The present method is amenable to synthesizing geared topologies" that include "linear-translation elements" (lines 205-206) should be better justified, as there is no relevant example of such topologies in the manuscript.

- The paragraph "To identify potential [...] with minimal rotation" (lines 207-211) is vague and unclear.

- The acknowledgments (p. 11) should be either completed or removed.

- On line 251, there is a reference to Fig. 9A, but there is only one Figure 9.

- The paragraph "the matrices cannot be [...] set as 205.35°" (lines 269-272) is vague and unclear. What is meant by "dramatize the motion" (line 277)? Notice that the mechanism in Fig. 2 can be driven by a "conventional DC motor" (line 278) even without the added dyad (links 6-7), which increases the mechanism complexity; the motion could then be simply limited to a defect-free range by controlling the motor rotation. Thus, the decision to add said dyad should be better justified.

## Conclusions

While the topic is interesting, the novelty and scope of this work appear limited. The bibliographic analysis ought to be expanded. I suggest to revise this work and to resubmit it at a more developed stage.

Comments on the Quality of English Language

Please see the rest of my review. The language used is overall acceptable and I recommend only minor edits for clarity.

Author Response

Thank you for your thoughtful feedback on the manuscript! It is clear in reading your response that you understand the subject matter and gave a good amount of time to your comments, and we appreciate that! We’ll address each of your concerns and suggestions here.

#General description

You make a good point that research in the field of geared linkages is vast beyond what we have referenced in this manuscript, and we appreciate your inclusion of several example works for us to cite. We have updated the references section including these articles, as well as several others that we have identified on our own. In reference to the high number of self-citations, Dr. Erdman, one of the authors on this paper, has been quite prolific in this field for many years. As a result, the references feel incomplete without including several of his previous works. With that said, we do recognize the percentage of these works is a bit high in the final reference total, and we have removed a few of the works which cite him as an author in order to bring the manuscript into alignment with the journal’s expectation of a maximum of 15% self-citation.

Other reviewers also noted that the conclusions section ought to be expanded and clarified, so we have taken some initiative to expand the section. Conclusions are then drawn in Sec. 3, and directions for future work are proposed, which are however too vague: what are the "optimal free choice values" and which kind of "defects" should be avoided (line 229)? I suggest to add a paragraph detailing the paper structure at the end of Sec. 1.

#Notes and Problems –

The scope of the work appears limited…

We agree with the reviewer’s assessment that the basis of the present method, mathematical synthesis using complex numbers, is not novel. However, we don’t view complex numbers as the primary contribution of the article. Rather, the article contributes a novel way of organizing geared linkage synthesis problems in which the mechanism is divided up into discrete chains. As we note in our literature review, we only identified one source that takes this approach to geared linkages, but their work was focused on analysis rather than synthesis, and they also used graph theory rather than complex numbers or other vector math. We have added a couple of statements attempting to clarify this primary purpose of the manuscript.

The manuscript is in a draft format and may have been submitted by mistake

You are correct, and we apologize for this error. The updated pdf version is free of these mistakes.

Style and organization:

The title has been updated to ‘segmentation’ instead of ‘dissolution’ to clarify the meaning.

The notation does not follow typographic notation:

We are curious which specific typographic notation the reviewer is referring to. We use the bold typeface intentionally in each of those equations to indicate that the quantity is a vector. This is critical for the reader’s understanding of the equations and is also consistent with many of the sources cited in this work. We do agree that symbols should be defined before they are introduced, and we have added some sentences to the body of the manuscript clarifying symbols in places where we noticed that we forgot to do this. δtriad in Figure 9 is indeed the same quantity as δ2 in Equation 2. We have chosen to use a different notation for the vector in the figure to help communicate the idea that the kinematic chains can be defined in this fashion regardless of their length (in terms of number of links). We’ll add a note to the Figure 9 caption to help make this point clearer to the readers.

The quality of the illustrations is not adequate for publication…

The specific figures addressed in the reviewer’s comment have been modified or explained in more detail in the text. A reduced version of Figure A2 was created which shows only 2 positions rather than 4, but it should be much easier for the reader to understand.

## Other minor issues

Thank you for taking the time to note these smaller issues in the manuscript, and for differentiating these comments from the rest of the comments in your review! Both aspects have made these modifications to our paper much more straightforward to complete.

Window casement mechanism – We acknowledge your point that the window casement mechanism is now quite old and have removed the claim that this is a ‘novel’ mechanism.

- Point B (fig 6) does not move along a "circular arc." Perhaps we should have been more precise with our language in this section, but our claim is true: When the link is rotating from position B to position B*, that point traces a circular arc with respect to point C. This is what creates the ‘dwell’ behavior of this particular special case mechanism. We’ve attempted to clarify the wording of this section.

- With respect to what is the "angular displacement" beta measured (line 164)? Some additional description has been added to this section to make each angular displacement’s meaning clearer.

- In Fig. 10, all kinematic chains have two Degrees-of-Freedom (DoFs): It makes sense that this was confusing, thank you for pointing this out! We intended for these chains to be incorporated into closed loop mechanisms, not as open loop chains. When the distal end is connected into a closed loop chain, or when the 2 variables values are prescribed in the problem, the chain falls to a single degree of freedom. This can be seen in the numerical example we provide in the appendix, where the furthest left chain from Figure 10 is implemented in a closed-loop chain with an ungeared dyad, and the overall system only has a single degree of freedom. We have clarified the wording of the text segment that you commented on in your initial review.

- Which position is considered in Tab. 1 (first column)? Is it the position of the end point P? The “Position” mentioned in the first column is the number of prescribed positions, not a particular position. We have clarified this wording.

- The last paragraph of p. 8 should be rewritten: what is meant by “free choice” here? We have added a statement addressing the idea of ‘free choices’ to this section.  

- Table 2 should not be split between separate pages; also, a table footnote (line 192) is present, but it is not clear to which part of the table it refers. Sorry for this confusion, we have more clearly identified which part of the table the footer is referring to. 

- The claim "The present method is amenable to synthesizing geared topologies" that include "linear-translation elements" (lines 205-206) should be better justified, as there is no relevant example of such topologies in the manuscript. Figure 12 depicts one simple example of a linear translating element incorporated into a geared linkage mechanism. We do not go out of our way to provide a numerical example of this concept, but we feel the addition is unnecessary and might clutter the core ideas of the paper. It is worth noting that in so called ‘basic kinematic diagrams,’ a common tool used to represent mechanism topologies, sliding elements are often represented as binary links. The discussion surrounding Figure 12 is intended to communicate this idea to the reader – any topological arrangement which contains a gear (or other f2 joint in kinematic diagram sense) should be amenable to replacing one of its binary links with a slider. In Figure 12, the sliding element is chosen as link 4, and the gear ratio is implemented between links 3 and 4.

- The paragraph "To identify potential [...] with minimal rotation" (lines 207-211) is vague and unclear. Taken along with the previous comment, we have added some additional description to this paragraph to make the idea clearer.

- The acknowledgments (p. 11) should be either completed or removed. We have removed the acknowledgements section.

- On line 251, there is a reference to Fig. 9A, but there is only one Figure 9. Thank you for pointing out this discrepancy, we have corrected the Figure identification.

- The paragraph "the matrices cannot be [...] set as 205.35°" (lines 269-272) is vague and unclear. What is meant by "dramatize the motion" (line 277)? Notice that the mechanism in Fig. 2 can be driven by a "conventional DC motor" (line 278) even without the added dyad (links 6-7), which increases the mechanism complexity; the motion could then be simply limited to a defect-free range by controlling the motor rotation. Thus, the decision to add said dyad should be better justified.

Some language has been added to the paragraph to clarify the meaning of those sentences. As for the dyad, we find it is almost always true that motors are one of, if not the most expensive part of designing a mechanism. Adjusting the build to accommodate a servo or stepper motor, along with the necessary control system and any required sensors, will dramatically drive up the cost of the electronics associated with this mechanism, and we would argue the risk of failure as well. Gaining the ability to drive a mechanism with a motor turning in a single direction is a tremendous benefit, and intrinsically eliminating any risk of entering a defect position by physically preventing the mechanism from entering these positions also reduces the risk of failure. We also find it valuable to add the driving dyad to help the reader understand that topologies can be modified and expanded even after a mechanism has been synthesized, especially through the addition of something as simple as a driving dyad which does not change the degrees of freedom. We hope this justification is sufficient to satisfy you. 

Round 2

Reviewer 2 Report

Comments and Suggestions for Authors

The authors  responded to my requirements.

Author Response

Thank you again for your feedback!

Reviewer 3 Report

Comments and Suggestions for Authors

After the first round of reviews, the authors have improved their manuscript and clarified it significantly; they have also provided a thoughtful and detailed rebuttal, which is very appreciated. However, some of the points which I previously raised are still valid. I thus recommend further revisions.

## Notes and problems

- The scope of the contribution still appears limited. While the treatment is technically correct, the methods are not novel, as the authors themselves acknowledge in the rebuttal. They state that the main contribution is "organizing geared linkage synthesis problems in which the mechanism is divided up into discrete chains", but this doesn't seem to add much beyond what can already be extrapolated from known works. I suggest to expand the example application in the Appendix (which could then become a Section in the main text), considering instead a practical design problem; this could be either derived from previous works, or a novel problem with a clear target application. Numerical data should be provided, as presently the output results in the Appendix are purely graphical. Preferably, a quantitative comparison could be provided with a non-geared, closed-loop, one-Degree-of-Freedom (1-DoF) design, such as a four-bar linkage, using the same target poses; this would help convincing the reader of the practical interest of geared linkages, which, as the authors observe, are generally more expensive than the alternatives.

- In the rebuttal, the authors state that they wanted to avoid cluttering their work with a detailed description of how their methods could be applied to linkages with rack-and-pinion couples. I understand their concern, but their claim at the end of p. 11 appears unjustified without at least a brief mathematical treatment.

## Style and organization

I agree with the authors that vectors should be bold, but subscripts (such as "2" in "delta_2") should not be bold. Symbols should be used coherently: while I understand the logic, denoting the same quantity as delta_triad in Fig. 10 and as delta_2 in Eq. (2) is confusing.

The quality of some figures has improved, yet Figs. 2, 3, 9, 11 and 12 are still blurry, with very small text that is hard to read. In Fig. A1, it is necessary to define T1 and so on (in the legend); the arrowheads here are unreadable. I advise against splitting tables across multiple pages (Tab. 1) or separating figures from their captions (Fig. 11).

## Other minor issues

- The work "A nine-bar linkage for mechanical forming presses" (Tso - Liang 2002), which I suggested in my previous review, has been disregarded. More broadly, the bibliography has been significantly revised and expanded, but mostly with works on general linkage design. I suggest this work could use some more references to example applications of planar geared linkages; the paper I mention by Tso and Liang is merely an example I found after a quick bibliographic search.

- Line 51: all positions are defined in a plane here, not in "space". I suspect hyphens are missing after "designer" and "problem", to connect these words to "defined".

- Line 67: the sentence "The approach" is unfinished.

- The authors observe in the rebuttal that adding a dyad reduces the risk of failure, but if the motor keeps "turning in a single direction" the linkage will get stuck against one of its limit positions, which actually increases said risk. Which link is driven by the motor in Fig. 2? If it is link 6, then the choice of adding a dyad would be better justified.

- Line 98: "quote" is a typo.

- Line 131: "f2" joints are nowhere defined in the manuscript.

- The observation that point B "traces a circular arc with respect to point C" (see rebuttal) is somewhat trivial, as points B and C are connected by a rigid link of length R. It would be clearer to state that the hypocycloid curve traced by B with respect to the fixed frame is close (up to the second order) to a circular arc of radius equal to R, which is what actually causes the dwell motion.

- Line 183: in "drive-up", the hyphen should be removed.

- Lines 188-189: what is meant by "dependent chains"? Are "complex number defined vectors" simply complex-valued vectors?

- I still suggest adding a paragraph detailing the manuscript structure at the end of Sec. 1. Since this section is quite long, I suggest to separate it into a general description of the topic and of this manuscript (Sec. 1) and a discussion of the state of the art (Sec. 2).

- Line 236: it should be specified that delta_j is "the positional displacement vector" of the j-th joint in the kinematic chain.

- The claim "Figure 10 reveals how Equation 2 might be reformulated for quadriad or dyad chains" is unconvincing; explicitly reporting the equivalent of Eq. (2) for the 2- or 4-link cases would be preferable.

- I now realize that, in Tabs. 1 and 2, it would be clearer to use "configurations" instead of "positions".

- The claim that "choosing a gear ratio [...] takes care of several free choice values" is a bit vague and incorrect in general (for example, if two angles related by a gear ratio are both unknown).

- The authors have answered in the rebuttal my observation on the number of DoFs of the linkages. However, how can the trace curve be uniquely defined by 2-DoF serial chains, even if these are connected to closed-loop chains? Notice that these latter chains are never taken into account in the analysis; at the very least, it should be specified that these are 1-DoF closed chains. The chains in Fig. 11 have two DoFs, except for the second one from the right, which has only one DoF: thus, that chain cannot be "incorporated" in a closed-loop chain, otherwise the result is a structure instead of a mechanism.

- The sentence "linear elements may present [...] to include them" is too vague.

- Lines 362-363 should in fact refer to the third part of Fig. 11 ("Link 2 to Link 3"). The figure suggests that the gear ratio is one to one, not "two to one".

- Equation (A3) should be written in radians, not in degrees. 

- Fusing two links (lines 393-394) changes the number of DoFs. It is still unclear to me how the mechanism can "achieve a larger sweeping arc" if prescribed positions are set. The reference to Fig. 2, which clearly shows a different mechanism, should be removed from the Appendix.

## Conclusions

While the topic is interesting, the novelty and scope of this work still appear somewhat limited. I suggest to revise this work and to expand its development and analysis.

Comments on the Quality of English Language  

Please see the rest of my review. The language used is fine and I recommend only minor edits for clarity.

Round 3

Reviewer 3 Report

Comments and Suggestions for Authors

## General description

After the second round of reviews, the authors have further improved their manuscript and added new material; again, I appreciate the careful rebuttal. I still recommend some revisions to clarify the manuscript: I leave to the Editors the decision on whether said revisions should be required for publication.

## Notes and problems

- I maintain that the description in lines 173-174 is inaccurate: point B does not move along a circular arc with respect to the frame, but instead follows a hypocycloid curve. It is true that this curve can be approximated by a circle of center C as the linkage moves around the configuration shown in Fig. 7.

- The authors have expanded their discussion of the contents of the manuscript at the end of Sec. 1. In my previous review, however, I suggested adding a description of the manuscript structure, as it is common for scientific papers in our field (e.g., "In Sec. 1, we discuss... In Sec. 2, we present... Finally, in ..."): I believe this would help the reader. I also suggest to divide Sec. 1, which is seven pages long, into a general description of the topic and of this manuscript (Sec. 1) and a discussion of the state of the art (Sec. 2). 

- After careful consideration, I still find that all kinematic chains in Fig. 12 have two DoFs, except for the second one, which has only one. Maybe the first gear should not be grounded? 

- In line 335, the authors state that they "implement a ratio of two to one between links two and three (As in Figure 11 “Ground to Link 2”)". This, however, is impossible, because link 3 is free to rotate around the rotary joint connecting it to link 2, and the angles of rotation of links 2 and 3 are unrelated.

- It is somewhat unclear whether the numerical values of the angles beta_2, beta_3 and beta_4 in Eq. (5) are arbitrarily chosen ("free  choices", line 347). Similarly, I do not understand whether the value beta_2=205.35° in line 364 was chosen arbitrarily.

- I still fail to understand how the mechanism in pages 13-15 can "achieve a larger sweeping arc motion" when its prescribed positions are fully set.

## Other minor issues

- Line 21: "the Antikythera" should be "the Antikythera mechanism" (Antikythera is the name of the island where said mechanism was recovered).

- Line 46: "a several example" is a typo.

- In Figs. 3a and 3b, it would be simpler to denote the planets as 2a, 2b and 2c, so the highlighted note on lines 100-101 can be removed. I also suggest removing the asterisks after 2 in Fig. 3a.

- What is meant by "dual arm version" in lines 106-107?

- I suggest reordering the discussion in lines 112-118 to follow the order of the figures (presenting first Fig. 4 and then Fig. 5).

- Some subscripts are still in boldface: see delta_j in line 255 or delta_2 in Eqs. (4) and (A1). Complex numbers also should not be in bold: see Eq. (5).

- I realize now that the naming of the angles in Fig. 10 is somewhat confusing: from the frame to the distal end, one finds angle beta, then alpha, then gamma. The order should be instead the same as the Greek alphabet (alpha, beta, gamma).

- On line 345, the authors refer to Eq. A2, which does not exist.

- Using the same symbols alpha, beta, W and Z for both Eqs. (5) and (6) is highly confusing, since they refer to different kinematic chains. Similarly, I suggest to change the notation in Figs. 14 and 15. In line 358, note that Eq. (6) gives three vector equations, not two.

- Labels (with the names of the variables and their measurement units) are needed for the axes in Figs. 16a and 17a.

- In Eq. (A1), is r a vector or a scalar? What is meant by "tail" of a chain (line 482)?

- Figure A2b is quite unclear. I suspect that several (partially overlapping) arrow heads are shown, relative to a very large arrow symbol; compare with Fig. A2a. If that is the case, I suggest avoiding arrows altogether and just using simple segments instead.

Comments on the Quality of English Language

I have no further comments to make; please see my review.
